# CONVEX REGULARIZATION BEHIND NEURAL RECONSTRUCTION

**Arda Sahiner, Morteza Mardani, Batu Ozturkler, Mert Pilanci, John Pauly**
Department of Electrical Engineering
Stanford University
{sahiner, morteza, ozt, pilanci, pauly}@stanford.edu

## ABSTRACT

Neural networks have shown tremendous potential for reconstructing high-resolution images in inverse problems. The non-convex and opaque nature of neural networks, however, hinders their utility in sensitive applications such as medical imaging. To cope with this challenge, this paper advocates a convex duality framework that makes a two-layer fully-convolutional ReLU denoising network amenable to convex optimization. The convex dual network not only offers the optimum training with convex solvers, but also facilitates interpreting training and prediction. In particular, it implies training neural networks with weight decay regularization induces path sparsity while the prediction is piecewise linear filtering. A range of experiments with MNIST and fastMRI datasets confirm the efficacy of the dual network optimization problem.

## 1   INTRODUCTION

In the age of AI, image reconstruction has witnessed a paradigm shift that impacts several applications ranging from natural image super-resolution to medical imaging. Compared with the traditional iterative algorithms, AI has delivered significant improvements in speed and image quality, making learned reconstruction based on neural networks widely adopted in clinical scanners and personal devices. The non-convex and opaque nature of deep neural networks however raises serious concerns about the authenticity of the predicted pixels in domains as sensitive as medical imaging. It is thus crucial to understand what the trained neural networks represent, and interpret their reconstruction per pixel for unseen images.

Reconstruction is typically cast as an inverse problem, where neural networks are used in different ways to create effective priors; see e.g., (Ongie et al., 2020; Mardani et al., 2018b) and references therein. An important class of methods are *denoising networks*, which given natural data corrupted by some noisy process $Y$, aim to regress the ground-truth, noise-free data $X_*$ (Gondara, 2016; Vincent et al., 2010). These networks are generally learned in a supervised fashion, such that a mapping $f : \mathcal{Y} \to \mathcal{X}$ is learned from inputs $\{y_i\}_{i=1}^n$ to outputs $\{x_{*i}\}_{i=1}^n$, and then can be used in the inference phase on new samples $\hat{y}$ to generate the prediction $\hat{x}_* = f(\hat{y})$.

The scope of supervised denoising networks is so general that it can cover more structured inverse problems appearing, for example, in compressed sensing. In this case one can easily form a poor (linear) estimate of the ground-truth image that is noisy and then reconstruct via end-to-end denoising networks (Mardani et al., 2018b; Mousavi et al., 2015). This method has been proven quite effective on tasks such as medical image reconstruction (Mardani et al., 2018b;a; Sandino et al., 2020; Hammernik et al., 2018), and significantly outperforms sparsity-inducing convex denoising methods, such as total-variation (TV) and wavelet regularization (Candès et al., 2006; Lustig et al., 2008; Donoho, 2006) in terms of both quality and speed.

Despite their encouraging results and growing use in clinical settings (Sandino et al., 2020; Hammernik et al., 2018; Mousavi et al., 2015), little work has explored the interpretation of supervised training of over-parameterized neural networks for inverse problems. Whereas robustness guarantees exist for inverse problems with minimization of convex sparsity-inducing objectives (Oymak & Hassibi, 2016; Chandrasekaran et al., 2012), there exist no such guarantees for predictions of

non-convex denoising neural networks based on supervised training. While the forward pass of a network has been interpreted as a layered basis pursuit from sparse dictionary learning, this approach lacks an understanding of the optimization perspective of such networks, neglecting the solutions to which these networks actually converge (Papyan et al., 2017). In fact, it has been demonstrated empirically that deep neural networks for image reconstruction can be unstable; i.e., small perturbations in the input can cause severe artifacts in the reconstruction, which can mask relevant structural features, which are important for medical image interpretation (Antun et al., 2020).

The main challenge in explaining these effects emanates from the non-linear and non-convex structure of deep neural networks that are heuristically optimized via first-order stochastic gradient descent (SGD) based solvers such as Adam (Kingma & Ba, 2014). As a result, it is hard to interpret the inference phase, and the training samples can alter the predictions for unseen images. In other applications, Neural Tangent Kernels (NTK) have become popular to understand the behavior of neural networks (Jacot et al., 2018). They however strongly rely on the oversimplifying *infinite-width* assumption for the network that is not practical, and as pointed out by prior work (Arora et al., 2019), they cannot explain the success of neural networks in practice. To cope with these challenges, we present a convex-duality framework for two-layer finite-width denoising networks with fully convolutional (conv.) layers with ReLU activation and the representation shared among all output pixels. In essence, inspired by the analysis by Pilanci & Ergen (2020), the zero-duality gap offers a convex bi-dual formulation for the original non-convex objective, that demands only polynomial variable count.

The benefits of the convex dual are three-fold. First, with the convex dual, one can leverage off-the-shelf convex solvers to guarantee convergence to the global optimum in polynomial time and provides robustness guarantees for reconstruction. Second, it provides an interpretation of the training with weight decay regularization as implicit regularization with path-sparsity, a form of capacity control of neural networks (Neyshabur et al., 2015). Third, the convex dual interprets CNN-based denoising as first dividing the input image patches into clusters, based on their latent representation, and then linear filtering is applied for patches in the same cluster. A range of experiments are performed with MNIST and fastMRI reconstruction that confirm the zero-duality gap, interpretability, and practicality of the convex formulation.

All in all, the main contributions of this paper are summarized as follows:

- We, for the first time, formulate a convex program with polynomial complexity for neural image reconstruction, which is provably identical to a two-layer fully-conv. ReLU network.

- We provide novel interpretations of the training objective with weight decay as path-sparsity regularization, and prediction as patch-based clustering and linear filtering.

- We present extensive experiments for MNIST and fastMRI reconstruction that our convex dual coincides with the non-convex neural network, and interpret the learned dual networks.

## 2 RELATED WORK

This paper is at the intersection of two lines of work, namely, convex neural networks, and deep learning for inverse problems. Convex neural networks were introduced in (Bach, 2017; Bengio et al., 2006), and later in (Pilanci & Ergen, 2020; Ergen & Pilanci, 2020a;b).

The most relevant to our work are (Pilanci & Ergen, 2020; Ergen & Pilanci, 2020b) which put forth a convex duality framework for two-layer ReLU networks with a single output. It presents a convex alternative in a higher dimensional space for the non-convex and finite-dimensional neural network. It is however restricted to *scalar-output* networks, and considers either fully-connected networks (Pilanci & Ergen, 2020), or, CNNs with average pooling (Ergen & Pilanci, 2020b). Our work however focuses on fully convolutional denoising with an output dimension as large as the number of image pixels, where these pixels share the same hidden representation. This is indeed quite different from the setting considered in (Pilanci & Ergen, 2020) and demands a different treatment. It could also be useful to mention that there are works in (Amos et al., 2017; Chen et al., 2019) that customize the network architecture for convex inference, but they still require non-convex training.

In recent years, deep learning has been widely deployed in inverse problems to either learn effective priors for iterative algorithms (Bora et al., 2017; Heckel & Hand, 2018), or to directly learn the inversion map using feed-forward networks (Jin et al., 2017; Zhang et al., 2017). In the former paradigm, either an input latent code, or, the parameters of a deep decoder network are optimized to generate a clean output image. A few attempts have been made to analyze such networks to explain their success (Yokota et al., 2019; Tachella et al., 2020; Jagatap & Hegde, 2019). Our work, in contrast, belongs to the latter group, which is of utmost interest in real-time applications, and thus widely adopted in medical image reconstruction. Compressed sensing (CS) MRI has been a successful fit, where knowing the forward acquisition model, one forms an initial linear estimate, and trains a non-linear CNNs to de-alias the input (Mardani et al., 2018a). Further, unrolled architectures inspired by convex optimization have been developed for robust de-aliasing (Sun et al., 2016; Mardani et al., 2018b; Hammernik et al., 2018; Sandino et al., 2020; Diamond et al., 2017). Past work however are all based on non-convex training of network filters, and interpretability is not their focus. Note that stability of iterative neural reconstructions has also been recently analyzed in Li et al. (2020); Mukherjee et al. (2020).

## 3 PRELIMINARIES AND PROBLEM STATEMENT

Consider the problem of denoising, i.e. reconstructing clean signals from ones which have been corrupted by noise. In particular, we are given a dataset of 2D [1] images $\boldsymbol{X}_* \in \mathbb{R}^{N \times h \times w}$, along with their corrupted counterparts $\boldsymbol{Y} = \boldsymbol{X}_* + \boldsymbol{E}$, where noise $\boldsymbol{E}$ has entries drawn from some probability distribution, such as $\mathcal{N}(0, \sigma^2)$ in the case of i.i.d. Gaussian noise. This is a fundamental problem, with a wide range of applications including medical imaging, image restoration, and image encryption problems (Jiang et al., 2018; Dong et al., 2018; Lan et al., 2019).

To solve the denoising problem, we deploy a two-layer CNN, where the first layer has an arbitrary kernel size $k$ and appropriately chosen padding, followed by an element-wise ReLU operation denoted by $(\cdot)_+$. The second and final layer of the network performs a conv. by a $1 \times 1$ kernel to generate the predictions of the network. The predictions generated by this neural network with $m$ first-layer conv. filters $\{\boldsymbol{u}_j\}_{j=1}^m$ and second-layer conv. filters $\{\boldsymbol{v}_j\}_{j=1}^m$ can be expressed as

$$f(\boldsymbol{Y}) = \sum_{j=1}^m (\boldsymbol{Y} \circledast \boldsymbol{u}_j)_+ \circledast \boldsymbol{v}_j \tag{1}$$

where $\circledast$ represents the 2D conv. operation.

### 3.1 TRAINING

We then seek to minimize the squared loss of the predictions of the network, along with an $\ell_2$-norm penalty (weight decay) on the network weights, to obtain the training problem

$$p^* = \min_{\substack{\boldsymbol{u}_j \in \mathbb{R}^{k \times k} \\ v_j \in \mathbb{R}}} \frac{1}{2} \| \sum_{j=1}^m (\boldsymbol{Y} \circledast \boldsymbol{u}_j)_+ \circledast v_j - \boldsymbol{X}_* \|_F^2 + \frac{\beta}{2} \sum_{j=1}^m \left( \|\boldsymbol{u}_j\|_F^2 + |v_j|^2 \right) \tag{2}$$

The network's output can also be understood in terms of matrix-vector products, when the input is appropriately expressed in terms of patch matrices $\{\boldsymbol{Y}_p \in \mathbb{R}^{k^2}\}_{p=1}^{Nhw}$, where each patch matrix corresponds to a patch of the image upon which a convolutional kernel will operate upon. Then, we can form the two-dimensional matrix input to the network as $\boldsymbol{Y}' = [\boldsymbol{Y}_1, \boldsymbol{Y}_2, \cdots, \boldsymbol{Y}_{Nhw}]^\top \in \mathbb{R}^{Nhw \times k^2}$, and attempt to regress labels $\boldsymbol{X}'_* \in \mathbb{R}^{Nhw}$, which is a flattened vector of the clean images $\boldsymbol{X}_*$. An equivalent form of the two-layer CNN training problem is thus given by

$$p^* = \min_{\substack{\boldsymbol{u}_j \in \mathbb{R}^{k^2} \\ v_j \in \mathbb{R}}} \frac{1}{2} \| \sum_{j=1}^m (\boldsymbol{Y}' \boldsymbol{u}_j)_+ v_j - \boldsymbol{X}'_* \|_2^2 + \frac{\beta}{2} \sum_{j=1}^m \left( \|\boldsymbol{u}_j\|_2^2 + |v_j|^2 \right) \tag{3}$$

In this form, the neural network training problem is equivalent to a 2-layer fully connected scalar-output ReLU network with $Nhw$ samples of dimension $k^2$, which has previously been theoretically analyzed (Pilanci & Ergen, 2020). We also note that for a fixed kernel-size $k$, the patch data matrix $\boldsymbol{Y}'$ has a fixed rank, since the rank of $\boldsymbol{Y}'$ cannot exceed the number of columns $k^2$.

---

[1] The results contained here are general to all types of conv., but we refer to the 2D case for simplicity.

## 3.2 ReLU Hyper-plane arrangements

To fully understand the convex formulation of the neural network proposed in (2), we must provide notation for understanding the hyper-plane arrangements of the network. We consider the set of diagonal matrices

$$\mathcal{D} := \{\mathrm{Diag}(\mathbf{1}_{\boldsymbol{Y}'\boldsymbol{u} \geq 0}) : \|\boldsymbol{u}\|_2 \leq 1\}$$

This set, which depends on $\boldsymbol{Y}'$, stores the set of activation patterns corresponding to the ReLU non-linearity, where a value of 1 indicates that the neuron is active, while 0 indicates that the neuron is inactive. In particular, we can enumerate the set of sign patterns as $\mathcal{D} = \{\boldsymbol{D}_i\}_{i=1}^{\ell}$, where $\ell$ depends on $\boldsymbol{Y}'$ and is bounded by

$$\ell \leq 2r \Big( \frac{e(Nhw - 1)}{r} \Big)^r$$

for $r := \mathbf{rank}(\boldsymbol{Y}')$ (Pilanci & Ergen, 2020). Thus, $\ell$ is polynomial in $Nhw$ for matrices with a fixed rank $r$, which occurs for convolutions with a fixed kernel size $k$. Using these sign patterns, we can completely characterize the range space of the first layer after the ReLU:

$$\{(\boldsymbol{Y}'\boldsymbol{u})_+ : \|\boldsymbol{u}\|_2 \leq 1\} = \{\boldsymbol{D}_i \boldsymbol{Y}'\boldsymbol{u} : \|u\|_2 \leq 1, \ (2\boldsymbol{D}_i - \boldsymbol{I})\boldsymbol{Y}'\boldsymbol{u} \geq 0, \ i \in [\ell]\}$$

With this notation established, we are ready to present our main theoretical result.

## 4 Convex duality

**Theorem 1.** *There exists an $m^* \leq Nhw$ such that if the number of conv. filters $m \geq m^* + 1$, the two-layer conv. network with ReLU activation (2) has a strong dual. This dual is a finite-dimensional convex program, given by*

$$p^* = d^* := \min_{\substack{(2\boldsymbol{D}_i - \boldsymbol{I})\boldsymbol{Y}'\boldsymbol{w}_i \geq 0 \\ (2\boldsymbol{D}_i - \boldsymbol{I})\boldsymbol{Y}'\boldsymbol{z}_i \geq 0}} \frac{1}{2} \| \sum_{i=1}^{\ell} \boldsymbol{D}_i \boldsymbol{Y}'(\boldsymbol{w}_i - \boldsymbol{z}_i) - \boldsymbol{X}'_*\|_2^2 + \beta \sum_{i=1}^{\ell} \Big( \|\boldsymbol{w}_i\|_2 + \|\boldsymbol{z}_i\|_2 \Big) \tag{4}$$

*where $\ell$ refers to the number of sign patterns associated with $\boldsymbol{Y}'$. Furthermore, given a set of optimal dual weights $(\boldsymbol{w}_i^*, \boldsymbol{z}_i^*)_{i=1}^{\ell}$, we can reconstruct the optimal primal weights as follows*

$$(\boldsymbol{u}_i^*, \boldsymbol{v}_i^*) = \begin{cases} (\frac{\boldsymbol{w}_i^*}{\sqrt{\|\boldsymbol{w}_i^*\|_2}}, \sqrt{\|\boldsymbol{w}_i^*\|_2}) & \boldsymbol{w}_i^* \neq 0 \\ (\frac{\boldsymbol{z}_i^*}{\sqrt{\|\boldsymbol{z}_i^*\|_2}}, \sqrt{\|\boldsymbol{z}_i^*\|_2}) & \boldsymbol{z}_i^* \neq 0 \end{cases} \tag{5}$$

It is useful to recognize that the convex program has $2\ell Nhw$ constraints and $2\ell k^2$ variables, which can be solved in polynomial time with respect to $N$, $h$ and $w$ using standard convex optimizers. For instance, using interior point solvers, in the worst case, the operation count is less than $\mathcal{O}\big(k^{12}(Nhw/k^2)^{3k^2}\big)$. Note also that our theoretical result contrasts with fully-connected networks analyzed in (Pilanci & Ergen, 2020), that demand exponential complexity in the dimension.

Our result can easily be extended to residual networks with skip connections as stated next.

**Corollary 1.1.** *Consider a residual two-layer network given by*

$$f_{res}(\boldsymbol{Y}) = \boldsymbol{Y} + \sum_{j=1}^{m} (\boldsymbol{Y} \circledast \boldsymbol{u}_j)_+ \circledast \boldsymbol{v}_j \tag{6}$$

*We can also pose the convex dual network (4) in a similar fashion, where now we simply regress upon the residual labels $\boldsymbol{X}_* - \boldsymbol{Y}$.*

### 4.1 Implicit regularization

In this section, we discuss the implicit regularization induced by the weight decay in the primal model (3). In particular, each dual variable $\boldsymbol{w}_i$ or $\boldsymbol{z}_i$ represents a *path* from the input to the output, since the the product of corresponding primal weights is given by

$$\boldsymbol{u}_i^* \boldsymbol{v}_i^* = \begin{cases} \boldsymbol{w}_i^* & \boldsymbol{w}_i^* \neq 0 \\ \boldsymbol{z}_i^* & \boldsymbol{z}_i^* \neq 0 \end{cases} \tag{7}$$

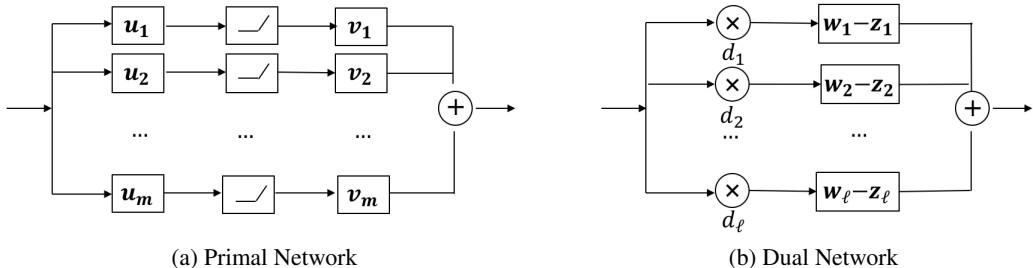

(a) Primal Network          (b) Dual Network

Figure 1: Primal and dual network interpretation. In the primal network, $m$ refers to the number of conv. filters, while in the dual network $\ell$ refers to the number of sign patterns.

Thus, the sparsity-inducing group-lasso penalty on the dual weights $w_i$ and $z_i$ induces sparsity in the paths of the primal model. In particular, a penalty is ascribed to $\|w_i\|_2 + \|z_i\|_2$, which in terms of primal weights corresponds to a penalty on $|v_i|\|u_i\|_2$. This sort of penalty has been explored previously in (Neyshabur et al., 2015), and refers to the path-based regularizer from their work.

## 4.2 INTERPRETABLE RECONSTRUCTION

The convex dual model (4) allows us to understand how an output pixel is predicted from a particular patch. Note that in this formulation, each input patch is regressed upon the center pixel of the output. In particular, for an input patch $y'_p$, the prediction of the network corresponding to the $p$-th output pixel is given by

$$f(y'_p) = \sum_{i=1}^{\ell} d_i^{(p)} y_p'^{\top} (w_i - z_i) \tag{8}$$

where $d_i^{(p)} \in \{0, 1\}$ refers to the $p$-th diagonal element of $D_i$. Thus, for an individual patch $y'_p$, the network can be interpreted as first selecting relevant sets of linear filters for that individual patch, and then taking a sum of the inner product of the patch with those filters–a piece-wise linear filtering operation. Thus, once it is identified which filters are active for a particular patch, the network's predictions are given as linear. This interpretation of the dual network contrasts with the opaque understanding of the primal network, in which due to the non-linear ReLU operation it is unclear how to interpret its predictions, as shown in Fig. 1.

Furthermore, because of the group-lasso penalty (Yuan & Lin, 2006) on $w_i$ and $z_i$ in the dual objective, these weights are sparse. Thus, for particular patch $y'_p$, only a few sign patterns $d_i^{(p)}$ influence its prediction. Therefore, different patches are implicitly clustered by the network according to the linear weights $w_i - z_i$ which are active for their predictions. A forward pass of the network can thus be considered as first a clustering operation, followed by a linear filtering operation for each individual cluster. As the neural network becomes deeper, we expect that this clustering becomes hierarchical–at each layer, the clusters become more complex, and capture more contextual information from surrounding patches.

## 4.3 DEEP NETWORKS

While the result of Theorem 1 holds only for two-layer fully conv. networks, these networks are essential for interpreting the implicit regularization and reconstruction of deeper neural networks. For one, these two-layer networks can be greedily trained to build a successively richer representation of the input. This allows for increased field of view for the piecewise linear filters to operate upon, along with allowing for more complex clustering of input patches. This approach is not dissimilar to the end-to-end denoising networks described by Mardani et al. (2018b), though it is more interpretable due to the simplicity of the convex dual of each successive trained layer.

This layer-wise training has been found to be successful in a variety of contexts. Greedily

pre-training denoising autoencoders layer-wise has been shown to improve classification performance in deep networks (Vincent et al., 2010). Greedy layer-wise supervised training has also been shown to perform competitively with much deeper end-to-end trained networks on image classification tasks (Belilovsky et al., 2019; Nøkland & Eidnes, 2019). Although analyzing the behavior of end-to-end trained deep networks is outside the scope of this work, we expect that end-to-end models are similar to networks trained greedily layerwise, which can fully be interpreted with our convex dual model.

# 5 EXPERIMENTS

## 5.1 MNIST DENOISING

**Dataset.** We use a subset of the MNIST handwritten digits (LeCun et al., 1998). In particular, for training, we select 600 gray-scale $28 \times 28$ images, pixel-wise normalized by the mean and standard deviation over the entire dataset. The full test dataset of 10,000 images is used for evaluation.

**Training.** We seek to solve the task of denoising the images from the MNIST dataset. In particular, we add i.i.d. noise from the distribution $\mathcal{N}(0, \sigma^2)$ for various noise levels, $\sigma \in \{0.25, 0.5, 0.75\}$. The resulting noisy images, $Y$, are the inputs to our network, and we attempt to learn the clean images $X_*$. We train both the primal network (2) and the dual network (4) using Adam (Kingma & Ba, 2014). For the primal network, we use 512 filters, whereas for the dual network, we randomly sample 8,000 sign patterns $D_i$ as an approximation to the full set of sign patterns $\ell$. Further experimental details can be found in the appendix.

**Zero duality gap.** We find that for this denoising problem, there is no gap between the primal and dual objective values across all values of $\sigma$ tested, verifying the theoretical result of Theorem 1, as demonstrated in Fig. 2. This is irrespective of the sign-pattern approximation, wherein we select only 8,000 sign patterns for the dual network, rather than enumerating the entire set of $\ell$ patterns. The illustrations of reconstructed images in Fig.3 also makes it clear that the primal and dual reconstructions are of similar quality.

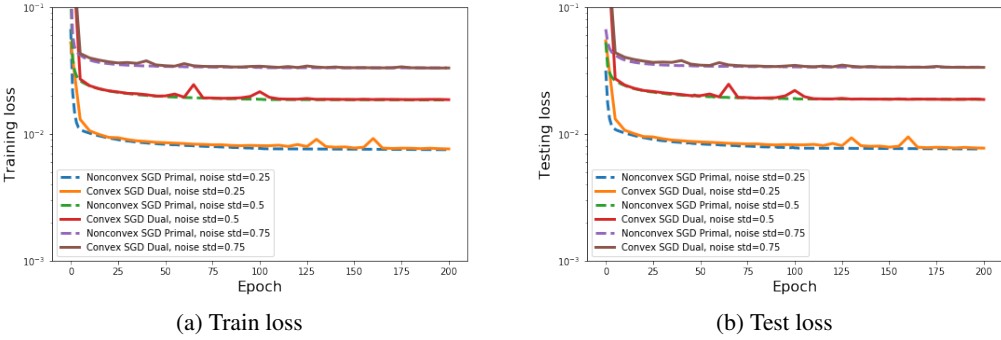

(a) Train loss         (b) Test loss

Figure 2: Train and test curves for MNIST denoising problem, for various noise levels $\sigma$.

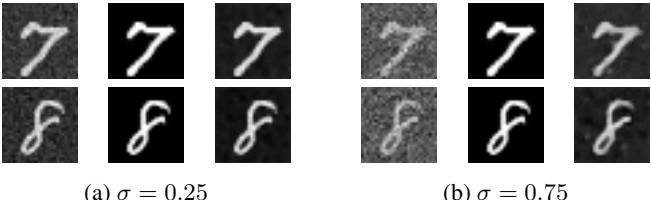

(a) $\sigma = 0.25$         (b) $\sigma = 0.75$

Figure 3: Test examples from MNIST denoising problem for two values of $\sigma$ from primal (top) and dual (bottom) networks. From left to right, images are: (a) noisy network input, (b) ground truth, (c) network output.

**Interpretable reconstruction.** Further, we can interpret what these networks have learned using our knowledge of the dual network. In particular, we can visualize both the sparsity of the learned filters, and the network's clustering of input patches. Because of the piece-wise linear nature of the dual network, we can visualize the dual filters $w_i$ or $z_i$ as linear filters for the selected sign patterns. Thus, the frequency response of these filters explains the filtering behavior of the end-to-end network, where depending on the input patch, different filters are activated. We visualize this frequency response of the dual weights $w_i$ in Fig. 4, where we randomly select 600 representative filters of size $28 \times 28$. We note that because of the path sparsity induced by the group-Lasso penalty on the dual weights, some of these dual filters are essentially null filters.

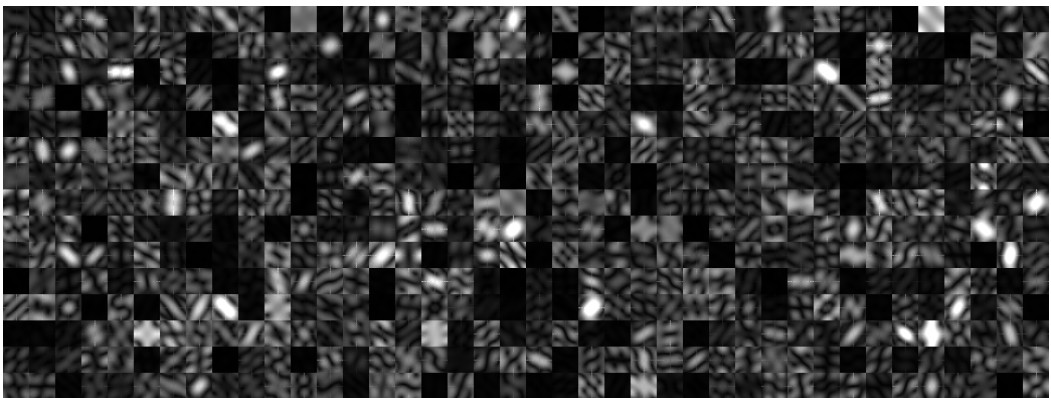

Figure 4: Visualization of the frequency response for the learned dual filters $\{w_i\}$ for denoising MNIST. Representative filters (600) are randomly selected for visualization when $\sigma = 0.5$.

The clustering of input patches can be detected via the set of sign patterns $d_i^{(p)}$ which correspond to non-zero filters for each output pixel $p$. Each output pixel $p$ can thus be represented by a binary vector $d^{(p)} \in \{0,1\}^\ell$. We thus feed the trained network clean test images and interpret how they are clustered, using k-means clustering with $k = 12$ to interpret the similarity among these binary vectors for each output pixel of an image. Visualizations of these clusters can be found in Fig. 5(a).

We can also use this clustering interpretation for deeper networks, even those trained end-to-end. We consider a four-layer architecture, which consists of two unrolled iterations of the two-layer architecture from the previous experiment, trained end-to-end. We can perform the same k-means clustering on the implicit representation obtained from each unrolled iteration, using the interpretation from the dual network. This result is demonstrated in Fig. 5(b), where we find that the clustering is more complex in the second iteration than the first, as expected. We note that while this network was trained end-to-end, the clusters from the first iteration are nearly identical to those of the single unrolled iteration, indicating that the early layers of end-to-end trained deeper denoising networks learn similar clusters to those of two-layer denoising networks.

## 5.2 MRI Reconstruction

**MRI acquisition.** In multi-coil MRI, the forward problem for each patient admits $y_i = \Omega F S_i x + e_i, i = 1, \cdots, n_c$ where $F$ is the 2D discrete Fourier transform, $\{S_i\}_{i=1}^{n_c}$ are the sensitivity maps of the receiver coils, and $\Omega$ is the undersampling mask that indexes the sampled Fourier coefficients.

**Dataset.** To assess the effectiveness of our method, we use the fastMRI dataset (Zbontar et al., 2018), a benchmark dataset for evaluating deep-learning based MRI reconstruction methods. We use a subset of the multi-coil knee measurements of the fastMRI training set that consists of 49 patients (1,741 slices) for training, and 10 patients (370 slices) for testing, where each slice is of size $80 \times 80$. We select $\Omega$ by generating Poisson-disc sampling masks using undersampling factors $R = 2, 4, 8$ with a calibration region of $16 \times 16$ using the SigPy python package (Ong & Lustig, 2019). Sensitivity maps $S_i$ are estimated using JSENSE (Ying & Sheng, 2007a).

**Training.** The multi-coil complex data are first undersampled, then reduced to a single-coil complex image using the SENSE model (Ying & Sheng, 2007b). The input of the networks are the real and

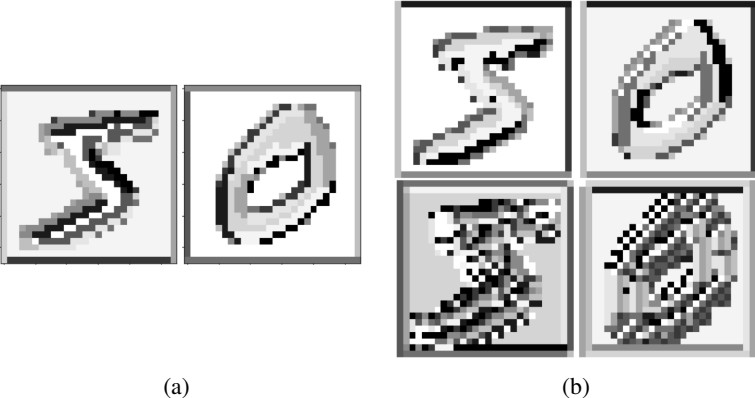

(a)                                                                (b)

Figure 5: Visualization of k-means clustering for latent representations of trained MNIST denoising network when $\sigma = 0.75$ and $k = 12$. (a) one unrolled iteration, (b) two unrolled iterations trained end to end; top row is the output of the first iteration, and bottom is the output of the second iteration.

imaginary components of this complex-valued Zero-Filled (ZF) image, where we wish to recover the fully-sampled ground-truth image. The real and imaginary components of each image are treated as separate examples during training. For the primal network, we use 1,024 filters, whereas for the dual network, we randomly sample 5,000 sign patterns.

**Zero duality gap.** We observe zero duality gap for CS-MRI, verifying Theorem 1. For different $R$, both the train and test loss of the primal and dual networks converge to the same optimal value, as depicted in Fig. 6. Furthermore, we show a representative axial slice from a random test patient in Fig. 7 reconstructed by the dual and primal networks, both achieving the same PSNR.

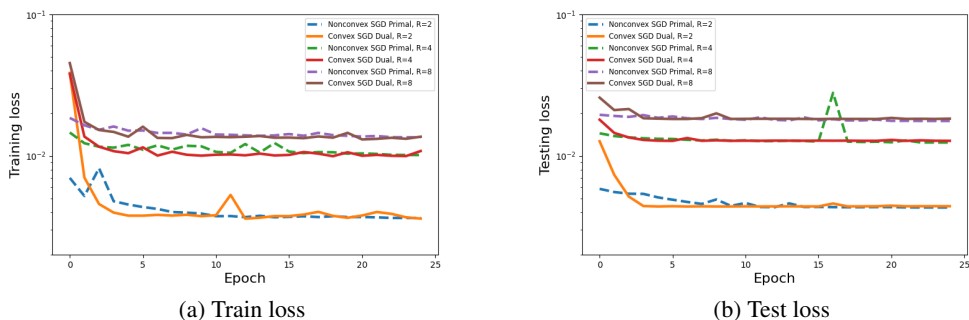

(a) Train loss                                                    (b) Test loss

Figure 6: Train and test curves for MRI reconstruction under various undersampling rates $R$.

## 6 CONCLUSIONS

This paper puts forth a convex duality framework for CNN-based denoising networks. Focusing on a two-layer CNN network with ReLU activation, a convex dual program is formulated that offers optimal training using convex solvers, and gains more interpretability. It reveals that the weight decay regularization of CNNs induces path sparsity regularization for training, while the prediction is piece-wise linear filtering. The utility of the convex formulation for deeper networks is also discussed using greedy unrolling. There are other important next directions to pursue. One such direction pertains to stability analysis of the convex neural network for denoising, and more extensive evaluations with pathological medical images to highlight the crucial role of convexity for robustness. Another such direction would be further exploration into fast and scalable solvers for the dual problem.

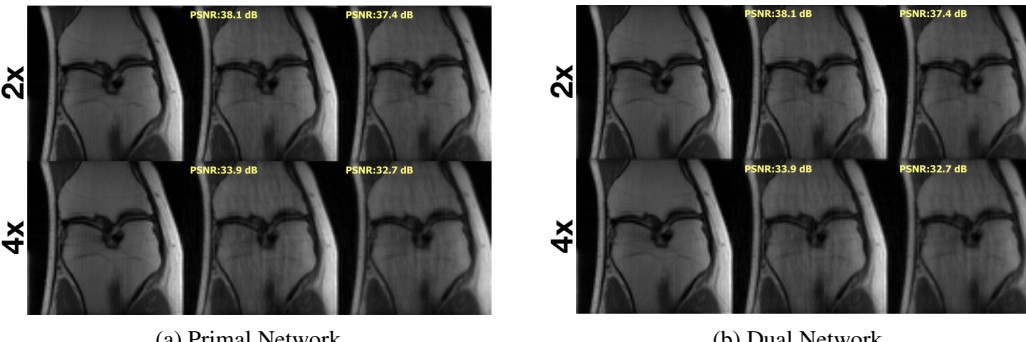

(a) Primal Network                               (b) Dual Network

Figure 7: Representative test knee MRI slice reconstructed via dual and primal network for undersampling $R = 2, 4$. From left to right: ground truth, output, and noisy ZF input.

## ACKNOWLEDGEMENTS

This work was partially supported by the National Science Foundation under grants IIS-1838179 and ECCS-2037304, the National Institutes of Health under grants R01EB009690 and R01EB0026136, Facebook Research, Adobe Research and Stanford SystemX Alliance.

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

## A  APPENDIX

## B  ADDITIONAL EXPERIMENTAL DETAILS

All experiments were run using the Pytorch deep learning library (Paszke et al., 2019). All primal networks are initialized using Kaiming uniform initialization (He et al., 2015). The losses presented are the sample and dimension-averaged, meaning that they differ by a constant factor of $Nhw$ from the formulas presented in (2) and (4). All networks were trained with an Adam optimizer, with $\beta_1 = 0.9$, $\beta_2 = 0.999$, and $\epsilon = 10^{-8}$. All networks had ReLU activation and had a final layer consisting of a $1 \times 1$ convolution kernel with stride 1 and no padding. For all dual networks, in order to enforce the feasibility constraints of (4), we penalize the constraint violation via hinge loss.

### B.1  ADDITIONAL EXPERIMENT: ROBUSTNESS FOR MNIST DENOISING

While for i.i.d. zero-mean Gaussian noise, training the primal and dual networks coincide, this may not necessarily be the case for all noise distributions. Depending on the distribution of noise, the loss landscape for the non-convex primal optimization problem may have more or fewer valleys–even if SGD does not get stuck in the an exact local minimum, it may converge extremely slowly, mimicking the effect of being stuck in a local minimum. In contrast, because the dual problem is convex, solving the dual problem is guaranteed to converge to the global minimum.

To see this effect, we compare the effect of denoising with i.i.d. zero-mean Gaussian noise, $\sigma = 0.75$, with that of i.i.d. exponentially distributed noise, with $\lambda = 1.15$. The exponential distribution has a higher noise variance, and also a heavier tailed distribution, and thus is more difficult for the non-convex problem to solve. For both cases, we employ 25 primal filters, and 8,000 sign patterns. The results of these experiments are shown in Figs 8 and 9. As we can see, for the Gaussian noise distribution, the primal and dual objective values coincide, but for the exponential noise distribution, the dual program performs better, suggesting the primal problem is stuck in a local minimum or valley. This suggests that when the data distribution is heavy-tailed, the convex dual network may be more robust to train than the non-convex primal network, due to a lack of local minima.

We train both the primal and the dual network in a distributed fashion on a NVIDIA GeForce GTX 1080 Ti GPU and NVIDIA Titan X GPU. For both cases, we use a kernel size of $3 \times 3$ with a unity stride and padding for the first layer. For the primal network, we train with a learning rate of $\mu = 10^{-1}$, whereas for the dual network we use a learning rate of $\mu = 10^{-3}$. We use a batch size of 25 for all cases. For the weight-decay parameter we use a value of $\beta = 10^{-5}$, which is not sample-averaged.

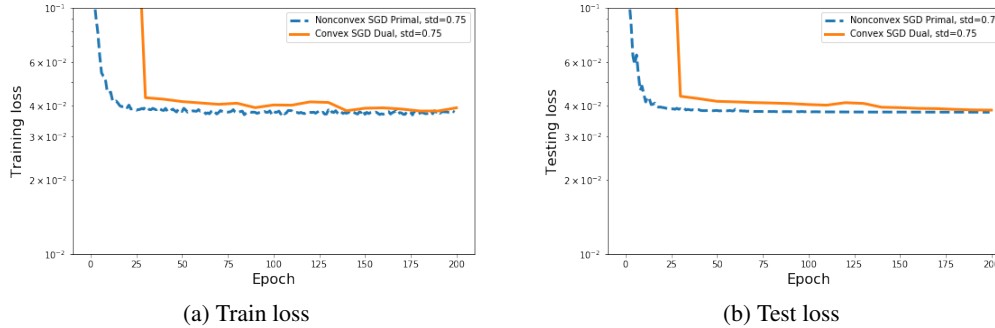

(a) Train loss              (b) Test loss

Figure 8: Train and test curves for Gaussian-distributed noise with $\sigma = 0.75$. The primal and dual optimization problems perform similarly well.

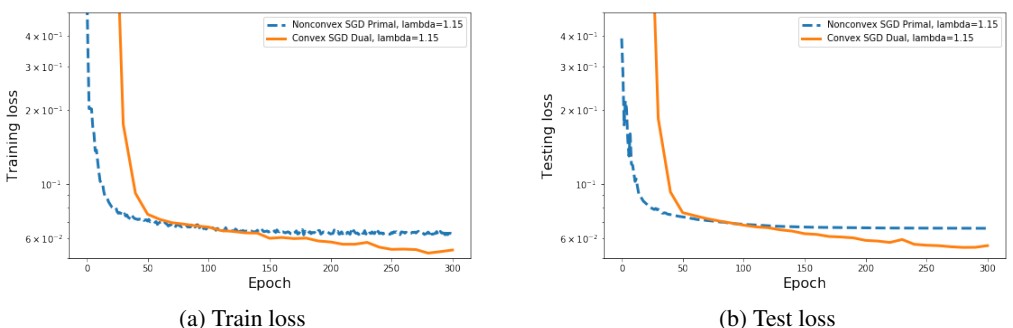

(a) Train loss              (b) Test loss

Figure 9: Train and test curves for exponentially distributed noise with $\lambda = 1.15$. The primal fails to learn as well as the dual.

### B.2 MNIST DENOISING

We train both the primal and the dual network in a distributed fashion on a NVIDIA GeForce GTX 1080 Ti GPU and NVIDIA Titan X GPU. For both cases, we use a kernel size of $3 \times 3$ with a unity stride and padding for the first layer. For the primal network, we train with a learning rate of $\mu = 10^{-3}$, whereas for the dual network we use a learning rate of $\mu = 10^{-5}$. We use a batch size of 25 for all cases. For the weight-decay parameter we use a value of $\beta = 10^{-5}$, which is not sample-averaged, and which is sufficiently large so as to prevent overfitting on the training set, as demonstrated in Fig. 2, which indicates that test performance is similar to training performance.

### B.3 ABLATION STUDY FOR THE NUMBER OF SIGN PATTERNS

For the MNIST denoising problem, we also considered the effect of changing the number of randomly sampled sign patterns to approximate the solution to the dual problem. The experimental setting of this ablation study is identical to the one of Section 5.1, namely, a primal network with 512 conv. filters, and additive i.i.d zero-mean Gaussian noise with variance $\sigma^2$. For the noise standard deviations of $\sigma = 0.5$ and $\sigma = 0.1$, we compare the test and training loss found by the primal network with that of the approximated dual network with a varied number of subsampled sign patterns.

Figures 10 summarizes our results. In particular, with higher amounts of noise, a larger number of sign patterns are required to be sampled in order to approximate the primal problem. With $\sigma = 0.5$, the dual problem requires approximately 4000 sign patterns to achieve zero duality gap, while with $\sigma = 0.1$, the dual problem requires only 125 sign patterns.

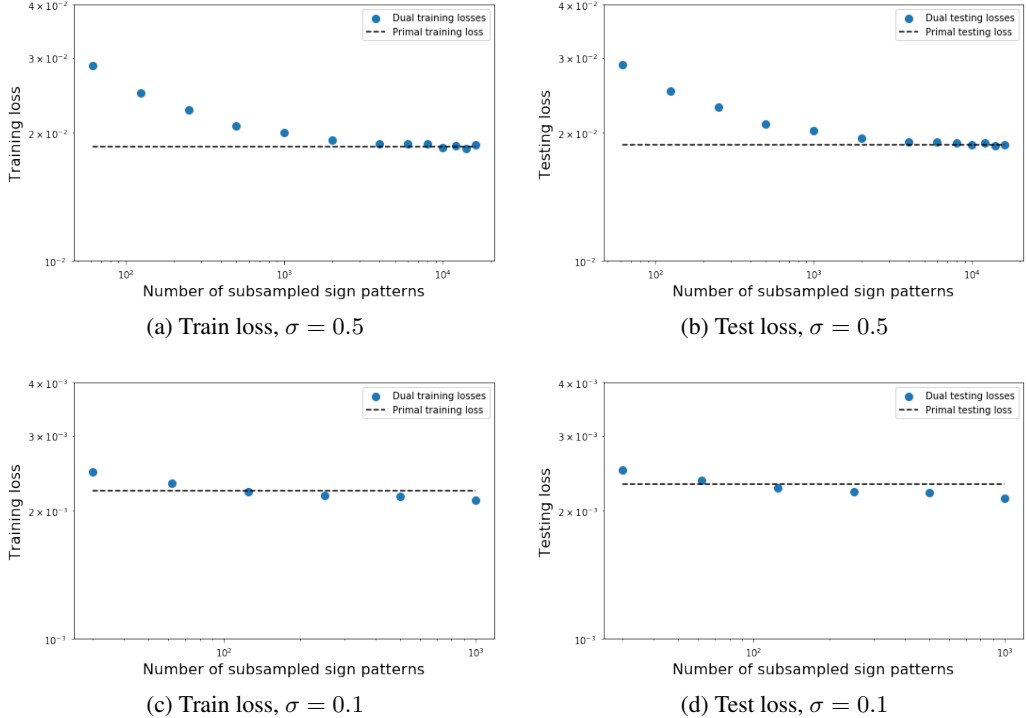

(a) Train loss, $\sigma = 0.5$            (b) Test loss, $\sigma = 0.5$

(c) Train loss, $\sigma = 0.1$            (d) Test loss, $\sigma = 0.1$

Figure 10: MNIST denoising with additive Gaussian noise and $\sigma \in \{0.5, 0.1\}$. Ablation study for the number of sampled sign patterns for the dual problem, compared to the primal problem with 512 filters.

### B.4 MRI Reconstruction

The fastMRI dataset has data acquired with $n_c = 15$ receiver coils. We train both the primal and the dual network on a single NVIDIA GeForce GTX 1080 Ti GPU with a batch size of 2. As our primal network architecture, we use a two-layer CNN with the first layer kernel size $7 \times 7$, unity stride and a padding of 3 with a skip connection. For the training of the primal and dual networks, networks are trained for 25 epochs. A learning rate of $\mu = 10^{-3}$ is used for the primal network, and a learning rate of $\mu = 5 \times 10^{-7}$ is used for the dual network with a weight-decay parameter of $\beta = 10^{-5}$.

### B.5 Learned Convolutional Filters for MRI Reconstruction

Similar to the visualized dual filters for MNIST reconstruction, we can also visualize the frequency response of the learned dual filters for the MRI reconstruction network, which we observe in Figure 11. We see that these filters select for a variety of frequencies, but are quite different visually from those for MNIST reconstruction.

## C Proof of Theorem 1

We note that, as mentioned in the main paper, (2) is equivalent to (3). Thus, to prove the theorem, it is sufficient to show that (3) is equivalent to (4). This proof follows directly from Theorem 1 from Pilanci & Ergen (2020).

Before delving into the details, the roadmap of the proof can be sketched as follows:

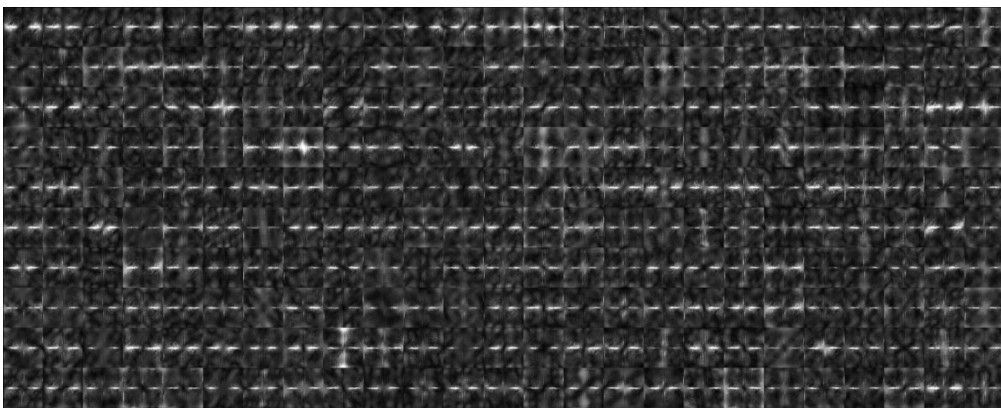

Figure 11: Visualization of the frequency response for the learned dual filters $\{\boldsymbol{w}_i\}$ for MRI reconstruction. Representative filters (250) of size $80 \times 80$ are randomly selected for visualization when $R = 4$.

1. Re-scale the original primal problem (3) to an equivalent form with $\ell_1$-regularization on the final layer weights

2. Use Lagrangian duality to eliminate the final layer weights and introduce a new dual variable to obtain an intractable, yet equivalent convex program.

3. Use Slater's condition to form the strong dual problem, which is a semi-infinite convex program.

4. Use the ReLU sign patterns $\{D_i\}_{i=1}^{\ell}$ to finitely parameterize this semi-infinite program, and then form the bi-dual to obtain the finite, convex, strong dual.

We begin by noting that (3) is equivalent to the following optimization problem:

$$p^* = \min_{\substack{\|\boldsymbol{u}_j\|_2 \leq 1 \\ v_j \in \mathbb{R}}} \frac{1}{2}\| \sum_{j=1}^{m}(\boldsymbol{Y}'\boldsymbol{u}_j)_+ v_j - \boldsymbol{X}'_*\|_2^2 + \beta \sum_{j=1}^{m} |v_j| \tag{9}$$

This is because, without changing the predictions of the network, we can re-scale $\boldsymbol{u}_j$ by some scalar $\gamma_j > 0$, granted that we also scale $v_j$ by its reciprocal $1/\gamma_j$. Then, we can simply minimize the regularization term over $\gamma_j$, noting that

$$\min_{\gamma_j} \|\gamma_j \boldsymbol{u}_j\|_2^2 + |v_j/\gamma_j|^2 = 2\|\boldsymbol{u}_j\|_2 |v_j| \tag{10}$$

From this we then obtain the equivalent convex program

$$p^* = \min_{\substack{\boldsymbol{u}_j \in \mathbb{R}^{k^2} \\ v_j \in \mathbb{R}}} \min_{\gamma_j \in \mathbb{R}} \frac{1}{2}\| \sum_{j=1}^{m}(\boldsymbol{Y}'\boldsymbol{u}_j)_+ v_j - \boldsymbol{X}'_*\|_2^2 + \beta \sum_{j=1}^{m} \|\gamma_j u_j\|_2^2 + |v_j/\gamma_j|^2 \tag{11}$$

$$= \min_{\substack{\boldsymbol{u}_j \in \mathbb{R}^{k^2} \\ v_j \in \mathbb{R}}} \min_{\gamma_j \in \mathbb{R}} \frac{1}{2}\| \sum_{j=1}^{m}(\boldsymbol{Y}'\boldsymbol{u}_j)_+ v_j - \boldsymbol{X}'_*\|_2^2 + \beta \sum_{j=1}^{m} \|u_j\|_2 |v_j| \tag{12}$$

Then, we can simply restrict $\|\boldsymbol{u}_j\|_2 \leq 1$ to obtain (9), noting that this does not change the optimal objective value. Then, from (9), we can form the equivalent problem via Langrangian duality. In particular, we can first re-parameterize the problem as

$$p^* = \min_{\|\boldsymbol{u}_j\|_2 \leq 1} \min_{v_j, \boldsymbol{r}} \frac{1}{2}\|\boldsymbol{r}\|_2^2 + \beta \sum_{j=1}^{m} |v_j| \text{ s.t. } \boldsymbol{r} = \sum_{j=1}^{m}(\boldsymbol{Y}'\boldsymbol{u}_j)_+ v_j - \boldsymbol{X}'_* \tag{13}$$

And then introduce the Lagrangian variable $\boldsymbol{z}$

$$p^* = \min_{\|\boldsymbol{u}_j\|_2 \leq 1} \min_{v_j, \boldsymbol{r}} \max_{\boldsymbol{z}} \frac{1}{2}\|\boldsymbol{r}\|_2^2 + \beta \sum_{j=1}^{m} |v_j| + \boldsymbol{z}^\top \boldsymbol{r} + \boldsymbol{z}^\top \boldsymbol{X}'_* - \boldsymbol{z}^\top \sum_{j=1}^{m}(\boldsymbol{Y}'\boldsymbol{u}_j)_+ v_j \tag{14}$$

Now, note that by Sion's minimax theorem, we can switch the inner maximum and minimum without changing the objective value, since the objective is convex in $v_j, \boldsymbol{r}$ and affine in $\boldsymbol{z}$, and obtain

$$p^* = \min_{\|\boldsymbol{u}_j\|_2 \leq 1} \max_{\boldsymbol{z}} \min_{v_j, \boldsymbol{r}} \frac{1}{2}\|\boldsymbol{r}\|_2^2 + \beta \sum_{j=1}^{m} |v_j| + \boldsymbol{z}^\top \boldsymbol{r} + \boldsymbol{z}^\top \boldsymbol{X}'_* - \boldsymbol{z}^\top \sum_{j=1}^{m} (\boldsymbol{Y}' \boldsymbol{u}_j)_+ v_j \quad (15)$$

Now, we compute the minimum over $\boldsymbol{r}$ to obtain

$$p^* = \min_{\|\boldsymbol{u}_j\|_2 \leq 1} \max_{\boldsymbol{z}} \min_{v_j} -\frac{1}{2}\|\boldsymbol{z}\|_2^2 + \beta \sum_{j=1}^{m} |v_j| + \boldsymbol{z}^\top \boldsymbol{X}'_* - \boldsymbol{z}^\top \sum_{j=1}^{m} (\boldsymbol{Y}' \boldsymbol{u}_j)_+ v_j \quad (16)$$

and then compute the minimum over $v_j$ to obtain

$$p^* = \min_{\|\boldsymbol{u}_j\|_2 \leq 1} \max_{|\boldsymbol{z}^\top (\boldsymbol{Y}' \boldsymbol{u}_j)_+| \leq \beta} -\frac{1}{2}\|\boldsymbol{z} - \boldsymbol{X}'_*\|_2^2 + \frac{1}{2}\|\boldsymbol{X}'_*\|_2^2 \quad (17)$$

We note that this semi-infinite program (17) is convex. As long as $\beta > 0$, this problem is strictly feasible (simply set $\boldsymbol{z} = 0$), hence by Slater's condition strong duality holds, and therefore we can form the dual problem

$$p^* = d^* := \max_{|\boldsymbol{z}^\top (\boldsymbol{Y}' \boldsymbol{u})_+| \leq \beta \ \forall \|\boldsymbol{u}\|_2 \leq 1} -\frac{1}{2}\|\boldsymbol{z} - \boldsymbol{X}'_*\|_2^2 + \frac{1}{2}\|\boldsymbol{X}'_*\|_2^2 \quad (18)$$

Now, this dual problem can be finitely parameterized using the sign patterns $\{\boldsymbol{D}_i\}_{i=1}^{\ell}$ using the pointwise maximum of the constraint. We thus have

$$d^* := \max_{\boldsymbol{z}} -\frac{1}{2}\|\boldsymbol{z} - \boldsymbol{X}'_*\|_2^2 + \frac{1}{2}\|\boldsymbol{X}'_*\|_2^2$$
$$\text{s.t.} \max_{\substack{i \in [\ell] \\ \|\boldsymbol{u}\|_2 \leq 1 \\ (2\boldsymbol{D}_i - \boldsymbol{I})\boldsymbol{Y}' \boldsymbol{u} \geq 0}} |\boldsymbol{z}^\top \boldsymbol{D}_i \boldsymbol{Y}' \boldsymbol{u}| \leq \beta \quad (19)$$

We can split this absolute value constraint into two constraints, and maximize in closed form over $\boldsymbol{u}$ by introducing Lagrangian variables $\alpha_i, \alpha'_i$:

$$d^* := \max_{\substack{\boldsymbol{z} \\ \alpha_i \geq 0 \\ \alpha'_i \geq 0}} -\frac{1}{2}\|\boldsymbol{z} - \boldsymbol{X}'_*\|_2^2 + \frac{1}{2}\|\boldsymbol{X}'_*\|_2^2$$
$$\text{s.t.} \ \|\boldsymbol{Y}'^\top (2\boldsymbol{D}_i - \boldsymbol{I})\alpha_i + \boldsymbol{Y}'^\top \boldsymbol{D}_i \boldsymbol{z}\| \leq \beta \ \forall i \in [\ell] \quad (20)$$
$$\|\boldsymbol{Y}'^\top (2\boldsymbol{D}_i - \boldsymbol{I})\alpha'_i - \boldsymbol{Y}'^\top \boldsymbol{D}_i \boldsymbol{z}\| \leq \beta \ \forall i \in [\ell]$$

We then form the Lagrangian

$$d^* := \max_{\substack{\boldsymbol{z} \\ \alpha_i \geq 0 \\ \alpha'_i \geq 0}} \min_{\lambda \geq 0, \lambda' \geq 0} -\frac{1}{2}\|\boldsymbol{z} - \boldsymbol{X}'_*\|_2^2 + \frac{1}{2}\|\boldsymbol{X}'_*\|_2^2$$
$$+ \sum_{i=1}^{\ell} \lambda_i \Big( \beta - \|\boldsymbol{Y}'^\top (2\boldsymbol{D}_i - \boldsymbol{I})\alpha_i + \boldsymbol{Y}'^\top \boldsymbol{D}_i \boldsymbol{z}\|_2 \Big) \quad (21)$$
$$+ \sum_{i=1}^{\ell} \lambda'_i \Big( \beta - \|\boldsymbol{Y}'^\top (2\boldsymbol{D}_i - \boldsymbol{I})\alpha'_i - \boldsymbol{Y}'^\top \boldsymbol{D}_i \boldsymbol{z}\|_2 \Big)$$

Noting by Sion's minimax theorem that strong duality holds, we can take the strong dual of the Lagrangian and not change the objective value. Flipping max and min, and maximizing with respect to $\boldsymbol{z}, \alpha_i$, and $\alpha'_i$ yields

$$d^* := \min_{\substack{\lambda \geq 0 \\ \lambda' \geq 0 \\ \|\boldsymbol{w}_i\|_2 \leq 1 \\ \|\boldsymbol{z}_i\|_2 \leq 1 \\ (2\boldsymbol{D}_i - \boldsymbol{I})\boldsymbol{Y}' \boldsymbol{w}_i \geq 0 \\ (2\boldsymbol{D}_i - \boldsymbol{I})\boldsymbol{Y}' \boldsymbol{z}_i \geq 0}} \frac{1}{2}\Big\|\Big( \sum_{i=1}^{\ell} \lambda_i \boldsymbol{D}_i \boldsymbol{Y}' \boldsymbol{w}_i - \lambda'_i \boldsymbol{D}_i \boldsymbol{Y}' \boldsymbol{z}_i \Big) - \boldsymbol{X}'_* \Big\|_2^2 + \beta \sum_{i=1}^{\ell} \lambda_i + \lambda'_i \quad (22)$$

We lastly note that we can perform a change of variables $\boldsymbol{w}_i := \lambda_i \boldsymbol{w}_i$ and $\boldsymbol{z}_i := \lambda'_i \boldsymbol{z}_i$, and then minimize over $\lambda_i$ and $\lambda'_i$ to obtain the final convex, finite-dimensional strong dual.

$$d^* := \min_{\substack{(2\boldsymbol{D}_i - \boldsymbol{I})\boldsymbol{Y}'\boldsymbol{w}_i \geq 0 \\ (2\boldsymbol{D}_i - \boldsymbol{I})\boldsymbol{Y}'\boldsymbol{z}_i \geq 0}} \frac{1}{2}\|\sum_{i=1}^{\ell} \boldsymbol{D}_i \boldsymbol{Y}'(\boldsymbol{w}_i - \boldsymbol{z}_i) - \boldsymbol{X}'_*\|_2^2 + \beta \sum_{i=1}^{\ell} \|\boldsymbol{w}_i\|_2 + \|\boldsymbol{z}_i\|_2 \tag{23}$$

Thus, we have that strong duality holds, i.e. $p^* = d^*$. By theory in semi-infinite programming, we know that $m^* + 1$ of the total $\ell$ filters $(\boldsymbol{w}_i, \boldsymbol{z}_i)$ are non-zero at optimum where $m^* \leq Nhw$ (Shapiro, 2009; Pilanci & Ergen, 2020).

Furthermore, we can use the relationship in (5) to also verify this result. In particular, note that in the general weak duality setting, we must have that $d^* \leq p^*$. Now, suppose we have optimal weights $\{(\boldsymbol{u}_i^*, \boldsymbol{v}_i^*)\}_{i=1}^{\ell}$ for the solution to the dual problem 4 obtaining objective value $d^*$. Then, using the relationship in (5), we can re-form the primal weights to (3), repeated here for convenience:

$$(\boldsymbol{u}_i^*, \boldsymbol{v}_i^*) = \begin{cases} (\frac{\boldsymbol{w}_i^*}{\sqrt{\|\boldsymbol{w}_i^*\|_2}}, \sqrt{\|\boldsymbol{w}_i^*\|_2}) & \boldsymbol{w}_i^* \neq 0 \\ (\frac{\boldsymbol{z}_i^*}{\sqrt{\|\boldsymbol{z}_i^*\|_2}}, \sqrt{\|\boldsymbol{z}_i^*\|_2}) & \boldsymbol{z}_i^* \neq 0 \end{cases} \tag{24}$$

Now, substituting these into the primal objective, we have

$$p^* = \min_{\substack{\boldsymbol{u}_j \in \mathbb{R}^{k^2} \\ v_j \in \mathbb{R}}} \frac{1}{2}\|\sum_{j=1}^{m} (\boldsymbol{Y}'\boldsymbol{u}_j)_+ v_j - \boldsymbol{X}'_*\|_2^2 + \frac{\beta}{2}\sum_{j=1}^{m}\left(\|\boldsymbol{u}_j\|_2^2 + |v_j|^2\right) \tag{25}$$

$$\leq \frac{1}{2}\|\sum_{i=1}^{\ell} (\boldsymbol{Y}'\boldsymbol{u}_i^*)_+ v_i - \boldsymbol{X}'_*\|_2^2 + \frac{\beta}{2}\sum_{i=1}^{\ell}\left(\|\boldsymbol{u}_i\|_2^2 + |v_i|^2\right) \tag{26}$$

$$= \frac{1}{2}\|\sum_{i=1}^{\ell} \boldsymbol{D}_i \boldsymbol{Y}'(\boldsymbol{w}_i - \boldsymbol{z}_i) - \boldsymbol{X}'_*\|_2^2 + \frac{\beta}{2}\sum_{i=1,\boldsymbol{w}_i^* \neq 0}^{\ell}\left(\|\frac{\boldsymbol{w}_i^*}{\sqrt{\|\boldsymbol{w}_i^*\|_2}}\|_2^2 + |\sqrt{\|\boldsymbol{w}_i^*\|_2}|^2\right) \tag{27}$$

$$+ \frac{\beta}{2}\sum_{i=1,\boldsymbol{z}_i^* \neq 0}^{\ell}\left(\|\frac{\boldsymbol{z}_i^*}{\sqrt{\|\boldsymbol{z}_i^*\|_2}}\|_2^2 + |\sqrt{\|\boldsymbol{z}_i^*\|_2}|^2\right) \tag{28}$$

$$= \frac{1}{2}\|\sum_{i=1}^{\ell} \boldsymbol{D}_i \boldsymbol{Y}'(\boldsymbol{w}_i^* - \boldsymbol{z}_i^*) - \boldsymbol{X}'_*\|_2^2 + \beta \sum_{i=1}^{\ell} \|\boldsymbol{w}_i^*\|_2 + \|\boldsymbol{z}_i^*\|_2 \tag{29}$$

$$= d^* \tag{30}$$

Thus, $p^* \leq d^*$. This combined with the weak duality result $d^* \leq p^*$ yields that $d^* = p^*$, as desired.

## D  PROOF OF COROLLARY 1

In this circumstance, we simply need to re-substitute the same objective as (4), with our new labels as $\boldsymbol{X}'_* - \boldsymbol{Y}_u$, where $\boldsymbol{Y}_u \in \mathbb{R}^{Nhw}$ is a flattened vector of the input image. Then, we simply have the problem given by

$$d^* = \min_{\substack{(2\boldsymbol{D}_i - \boldsymbol{I})\boldsymbol{Y}'\boldsymbol{w}_i \geq 0 \\ (2\boldsymbol{D}_i - \boldsymbol{I})\boldsymbol{Y}'\boldsymbol{z}_i \geq 0}} \|\sum_{i=1}^{P} \boldsymbol{D}_i \boldsymbol{Y}'(\boldsymbol{w}_i - \boldsymbol{z}_i) - (\boldsymbol{X}'_* - \boldsymbol{Y}_u)\|_2^2 + \beta \sum_{i=1}^{P}\left(\|\boldsymbol{w}_i\|_2 + \|\boldsymbol{z}_i\|_2\right) \tag{31}$$

Thus, the general form of the convex dual formulation still holds with a simple residual network.

# E    EXTENSION TO GENERAL CONVEX LOSS FUNCTIONS

The results of Theorem 1 and Corollary 1 can be extended to any arbitrary convex loss function $L(\hat{\boldsymbol{X}}, \boldsymbol{X}_*)$. In particular, consider the non-convex primal training problem

$$p^* := \min_{\substack{\boldsymbol{u}_j \in \mathbb{R}^{k^2} \\ v_j \in \mathbb{R}}} L\Big(\sum_{j=1}^{m}(\boldsymbol{Y}'\boldsymbol{u}_j)_+ v_j, \boldsymbol{X}_*'\Big) + \frac{\beta}{2}\sum_{j=1}^{m}\Big(\|\boldsymbol{u}_j\|_2^2 + |v_j|^2\Big) \tag{32}$$

for $m \geq m^*$ as defined previously. Then, this problem has a convex strong bi-dual, given by

$$p^* = d^* := \min_{\substack{(2\boldsymbol{D}_i - \boldsymbol{I})\boldsymbol{Y}'\boldsymbol{w}_i \geq 0 \\ (2\boldsymbol{D}_i - \boldsymbol{I})\boldsymbol{Y}'\boldsymbol{z}_i \geq 0}} L\Big(\sum_{i=1}^{\ell} \boldsymbol{D}_i \boldsymbol{Y}'(\boldsymbol{w}_i - \boldsymbol{z}_i), \boldsymbol{X}_*'\Big) + \beta\sum_{i=1}^{\ell}\Big(\|\boldsymbol{w}_i\|_2 + \|\boldsymbol{z}_i\|_2\Big) \tag{33}$$

The proof for this strong dual is almost identical to that in Appendix C. In particular, we first define the Fenchel conjugate function

$$L^*(\boldsymbol{z}) = \max_{\boldsymbol{r}} \boldsymbol{r}^\top \boldsymbol{z} - L(\boldsymbol{z}, \boldsymbol{X}_*') \tag{34}$$

Now, note that we can re-write (32) in re-scaled form as in (9):

$$p^* = \min_{\substack{\|\boldsymbol{u}_j\|_2 \leq 1 \\ v_j \in \mathbb{R}}} L\Big(\sum_{j=1}^{m}(\boldsymbol{Y}'\boldsymbol{u}_j)_+ v_j, \boldsymbol{X}_*'\Big) + \beta\sum_{j=1}^{m}|v_j| \tag{35}$$

The strong dual of (35) is then given by

$$p^* = d^* := \max_{\boldsymbol{z}} -L^*(\boldsymbol{z}) \text{ s.t. } |\boldsymbol{z}^\top(\boldsymbol{Y}'\boldsymbol{u})_+| \leq \beta \; \forall \|\boldsymbol{u}\|_2 \leq 1 \tag{36}$$

using standard Fenchel duality (Boyd et al., 2004). We can further re-write the constraint set in terms of sign patterns as done in the proof in Appendix C. Then, we can follow the same steps from the proof of Theorem 1, noting that by Fenchel–Moreau Theorem, $L^{**} = L$ (Borwein & Lewis, 2010). Thus, we obtain the convex-bi dual

$$p^* = d^* := \min_{\substack{(2\boldsymbol{D}_i - \boldsymbol{I})\boldsymbol{Y}'\boldsymbol{w}_i \geq 0 \\ (2\boldsymbol{D}_i - \boldsymbol{I})\boldsymbol{Y}'\boldsymbol{z}_i \geq 0}} L\Big(\sum_{i=1}^{\ell} \boldsymbol{D}_i \boldsymbol{Y}'(\boldsymbol{w}_i - \boldsymbol{z}_i), \boldsymbol{X}_*'\Big) + \beta\sum_{i=1}^{\ell}\Big(\|\boldsymbol{w}_i\|_2 + \|\boldsymbol{z}_i\|_2\Big) \tag{37}$$

as desired.

# F    FAILURE OF STRAIGHTFORWARD DUALITY ANALYSIS FOR OBTAINING A TRACTABLE CONVEX DUAL

In this section, we discuss how straightforward duality analysis will fail to generate a tractactable convex dual formulation of the two-layer ReLU-activation fully-conv. network training problem. In particular, we will follow an alternative proof to that in Appendix C, omitting the re-scaling step, and demonstrate that the dual becomes intractable.

We begin with (3)

$$p^* = \min_{\substack{\boldsymbol{u}_j \in \mathbb{R}^{k^2} \\ v_j \in \mathbb{R}}} \frac{1}{2}\|\sum_{j=1}^{m}(\boldsymbol{Y}'\boldsymbol{u}_j)_+ v_j - \boldsymbol{X}_*'\|_2^2 + \frac{\beta}{2}\sum_{j=1}^{m}\Big(\|\boldsymbol{u}_j\|_2^2 + |v_j|^2\Big) \tag{38}$$

We can re-parameterize the problem as

$$p^* = \min_{\substack{\boldsymbol{u}_j \in \mathbb{R}^{k^2} \\ v_j \in \mathbb{R}}} \min_{\boldsymbol{r}} \frac{1}{2}\|\boldsymbol{r}\|_2^2 + \frac{\beta}{2}\sum_{j=1}^{m}\Big(\|\boldsymbol{u}_j\|_2^2 + |v_j|^2\Big) \text{ s.t. } \boldsymbol{r} = \sum_{j=1}^{m}(\boldsymbol{Y}'\boldsymbol{u}_j)_+ v_j - \boldsymbol{X}_*' \tag{39}$$

And then introduce the Lagrangian variable $\boldsymbol{z}$

$$p^* = \min_{\substack{\boldsymbol{u}_j \in \mathbb{R}^{k^2} \\ v_j \in \mathbb{R}}} \min_{\boldsymbol{r}} \max_{\boldsymbol{z}} \frac{1}{2}\|\boldsymbol{r}\|_2^2 + \frac{\beta}{2}\sum_{j=1}^m \left(\|\boldsymbol{u}_j\|_2^2 + |v_j|^2\right) + \boldsymbol{z}^\top \boldsymbol{r} + \boldsymbol{z}^\top \boldsymbol{X}'_* - \boldsymbol{z}^\top \sum_{j=1}^m (\boldsymbol{Y}'\boldsymbol{u}_j)_+ v_j \quad (40)$$

Now, we can use Sion's minimax theorem to exchange the maximum over $\boldsymbol{z}$ and the minimums over $\boldsymbol{r}$ and $v_j$.

$$p^* = \min_{\boldsymbol{u}_j} \max_{\boldsymbol{z}} \min_{\boldsymbol{r}, v_j} \frac{1}{2}\|\boldsymbol{r}\|_2^2 + \frac{\beta}{2}\sum_{j=1}^m \left(\|\boldsymbol{u}_j\|_2^2 + |v_j|^2\right) + \boldsymbol{z}^\top \boldsymbol{r} + \boldsymbol{z}^\top \boldsymbol{X}'_* - \boldsymbol{z}^\top \sum_{j=1}^m (\boldsymbol{Y}'\boldsymbol{u}_j)_+ v_j \quad (41)$$

Minimizing over $\boldsymbol{r}$, we obtain

$$p^* = \min_{\boldsymbol{u}_j} \max_{\boldsymbol{z}} \min_{v_j} -\frac{1}{2}\|\boldsymbol{z}\|_2^2 + \frac{\beta}{2}\sum_{j=1}^m \left(\|\boldsymbol{u}_j\|_2^2 + |v_j|^2\right) + \boldsymbol{z}^\top \boldsymbol{X}'_* - \boldsymbol{z}^\top \sum_{j=1}^m (\boldsymbol{Y}'\boldsymbol{u}_j)_+ v_j \quad (42)$$

Now, minimizing over $v_j$, we obtain the optimality condition that $v_j^* = \frac{1}{\beta}\boldsymbol{z}^\top(\boldsymbol{Y}'\boldsymbol{u}_j)_+ \ \forall j$. Re-substituting this, we obtain the problem

$$p^* = \min_{\boldsymbol{u}_j} \max_{\boldsymbol{z}} -\frac{1}{2}\|\boldsymbol{z} - \boldsymbol{X}'_*\|_2^2 + \frac{1}{2}\|\boldsymbol{X}'_*\|_2^2 + \frac{\beta}{2}\sum_{j=1}^m \|\boldsymbol{u}_j\|_2^2 + \frac{\beta-2}{2\beta}\sum_{j=1}^m \left(\boldsymbol{z}^\top(\boldsymbol{Y}'\boldsymbol{u}_j)_+\right)^2 \quad (43)$$

This problem is not convex, hence strong duality does not hold (we cannot switch max and min without changing the objective), and we cannot maximize over $\boldsymbol{z}$ in closed form since the optimiality condition for $\boldsymbol{z}$ is given as

$$\boldsymbol{z} - \boldsymbol{X}'_* = \frac{\beta-2}{\beta}\sum_{j=1}^m \left(\boldsymbol{z}^\top(\boldsymbol{Y}'\boldsymbol{u}_j)_+\right)(\boldsymbol{Y}'\boldsymbol{u}_j)_+$$

Thus, standard duality approaches fail to find a tractable dual to the neural network training problem (3). In particular, the re-scaling step (9) and the introduction of sign patterns (19) are detrimental to render our analysis tractable.

