# OpenReview forum: "Convex Regularization behind Neural Reconstruction"
_ICLR.cc/2021/Conference — ICLR 2021 Poster_

### Official Review · AnonReviewer2 · 2020-10-23
**Interesting but not enough discussion with related works**

**Rating:** 6
**Confidence:** 4

**Review:**

This paper describes the interpretation of image restoration by fully convolutional neural network (FCNN) through the dual problem of neural network learning. Since the activation function is non-linear, it is generally difficult to interpret the main problem. In this paper, Authors consider a huge pattern of activation as samples and define the corresponding filter as dual parameters. As a result, when a finite number of sufficiently large filters is prepared, this becomes a convex dual problem of the primal problem. This dual framework itself is the work in [Pilanci & Ergen, ICML2020], and it is considered that the actual contribution of this paper was to apply it to the image resconstruction problem and discuss its interpretation.

The conclusions about the interpretation of FCNN were that the l2 penalty of the weight parameters was group lasso in the dual parameter and sparse regularization of the path, and that denoising was patch-based clustering and its linear filtering.

The interpretation of neural image restoration through the dual problem is very interesting and I think it is well written as a paper. It is nice that an application example of medical image restoration is also shown.

However, there are some recent studies that are considered to be very closely related to this paper, and the novelty of the conclusion itself of patch-based image processing seems to be weak. The papers that are considered to be closely related are listed below.

 --Convolutional sparse coding
Papyan, Vardan, Yaniv Romano, and Michael Elad. "Convolutional neural networks analyzed via convolutional sparse coding." The Journal of Machine Learning Research 18.1 (2017): 2887-2938.
**short summary** In the above paper, the CNN filters are interpreted as a patch dictionary, and the sparse feature map is interpreted as a coefficient. Image reconstruction by CNN is interpreted as patch-based sparse coding.

 --Low-dimensional manifold modeling of patches in a single image
Yokota, Tatsuya, et al. "Manifold Modeling in Embedded Space: A Perspective for Interpreting Deep Image Prior." ArXiv preprint arXiv: 1908.02995 (2019).
**short summary** In the above paper, the operation of extracting many patches from a single image is considered as embedding, and the low-dimensional manifold of patches embedded in the $k^2$-dimensional Euclidean space is learned by denoising auto-encoder. Image reconstruction by CNN is interpreted as low-dimensional manifold learning of patches.

 --Neural tangent denoiser
Tachella, Julián, Junqi Tang, and Mike Davies. "CNN Denoisers As Non-Local Filters: The Neural Tangent Denoiser." ArXiv preprint arXiv: 2006.02379 (2020).
**short summary** In the above paper, the noise removal by CNN is interpreted using the theory of neural tangent kernel, which has been attracting attention in recent years. The theory of neural tangent kernel shows that CNN can be interpreted as filtering using the similarity matrix between patches when the number of filters becomes very large. I feel that there is a relationship between increasing the number of filters in the above paper and increasing $ l $ in the dual representation of this paper.

I think it would be even better if the relationships between this paper and above papers are discussed in the manuscript.

---

> ### Author Response · Authors · 2020-11-17
> **Response to Reviewer 2**
>
> We would like to thank the reviewer for the comments as well as pointing out the new references. We have cited the new references, and discussed their difference from our work in the related work section of the revised manuscript. We should however note that they are quite different from ours in several ways. The main differences are summarized below:
>
> - **[Papyan et al’2017] -- Sparse Coding**
>
> Thank you for bringing this work to our attention. This work is quite interesting for understanding CNNs for signal reconstruction. The authors relate CNNs to traditional sparse coding, by interpreting the forward pass of a CNN as a layered basis-pursuit algorithm. Unlike ours, this work does not consider the effect of CNN training but instead just models its forward pass. This work thus does not focus on the task of analyzing or interpreting the optimization process of a neural network, only its predictions. We have cited this in the revised paper and distinguished it from our work.
>
> - **[Yokota et al’2019] - Low-dimensional manifold modeling of patches in a single image**
>
> Thank you for mentioning this work. The setting in this work is slightly different from ours. In particular, it follows the style of Deep Image Prior (DIP) works, which explore the setting of denoising a single noisy image by generating an image from a CNN and attempting to minimize the loss between the generated image and the noisy image. In contrast, in our setting, we train CNNs with noisy images as inputs and outputs are the ground truth images, and we learn a mapping from noisy images to ground truth images.
>
> As you have mentioned, the motivations and basic intuition behind the work are quite similar to ours, in that they interpret CNNs as a low-dimensional manifold learning of patches. As such, in our revised version, we have added additional discussion on DIP works, including Yokota et al (2019), and remark on the differences.
>
> - **[Tachella et al’2020] -- Neural tangent denoiser**
>
> Similar to [Yokota et al’2019], this work examines neural networks in the context of DIP--which is a different setting from ours. Beyond this, the Neural Tangent Kernel (NTK) has also been used to analyze neural networks, but that is only useful in the infinite-width limit, and does not apply for finite-width neural networks as they are used in practice.
>
> Moreover, NTK analysis will not explain the success of practical and finite-width neural networks (see for example Table 1 of [Arora et al, NeurIPS ’19]). [Tachella et al’20] even mentions that networks trained with Adam outperform the NTK solution for these DIP problems. We have cited this in the revised paper, and also have distinguished our work from the NTK.
>
> **References**
> Sanjeev  Arora,  Simon  S  Du,  Wei  Hu,  Zhiyuan  Li,  Russ  R  Salakhutdinov,  and  Ruosong  Wang. On exact computation with an infinitely wide neural net. *In Advances in Neural Information Processing Systems*, pp. 8141–8150, 2019.

---

### Official Review · AnonReviewer4 · 2020-10-28

**Rating:** 9
**Confidence:** 5

**Review:**

Summary and Contributions: This paper aims to find convex alternatives for deep learning based image reconstruction problems. This is well motivated by medical imaging, where there is risk of hallucinating unseen pixels, and there is high demand for robustness of training and interpretability of the prediction. For a two-layer convolutional neural network with ReLU activation, with weight-decay regularization, that is non-convex, this paper establishes strong convex duality, where the dual optimization is quite tractable and interpretable. It also shows empirical results for MNIST denoising and MRI reconstruction that support the claims.

Overall, I enjoyed reading this paper. It is a solid work, very well written, and it has a balanced mix of theory and application, where the theory shows direct practical impact.

Strong points:
-   This paper is very well written and well organized. The motivation is very clear and sound.
-   The proposed convex duality framework is solid and it fits nicely an important application with minimal assumptions.
-   The interpretability offered by the dual network is very interesting. This is a novel and elegant view. In particular, the filters visualized in Fig. 4 are insightful to what a neural network learns. The clustering interpretation is also neat.
-  The experiments with MNIST and fastMRI are convincing. In particular, the experiments with MRI are important and practically valuable, since in medical imaging deep learning has become the standard method nowadays for reconstruction, and, no or little is known about the interpretability of existing methods.

Additional feedback and suggestions:
- The appendix (Fig. 9) includes experiments with impulsive noise that show SGD for convex dual network converges to a better training/validation loss than the non-convex network. It might be better to move this to the main paper, as this is the message that the paper is trying to convey?
- Typos and notation mismatch in the proof of Theorem 1 in the appendix: what is X in eq. 13? index j in eq. 12?
- In the final version, it would be also useful to add the filter visualization for the MRI reconstruction as well. It would be interesting to see how dataset dependent the learned filters are.
- I am wondering about the complexity of the dual optimization problem. Are there fast convex solvers for scaling up the computations? Also, about the required number of sign patterns, can the authors provide intuitions for the datasets used in the paper, say for MNIST?

Correctness: The analyses are solid and the claims are correct.
Relation to prior work: The paper has clearly discussed the connections with the previous works.
Reproducibility: The details of the experiments and the public datasets are included in the paper. Thanks also for sharing the code.

---

> ### Author Response · Authors · 2020-11-17
> **Response to Reviewer 4**
>
> We would like to thank the reviewer for their helpful review and positive opinion. We have addressed all the reviewer comments below.
>
> - **"The appendix (Fig. 9) includes experiments with impulsive noise that show SGD for convex dual network converges to a better training/validation loss than the non-convex network. It might be better to move this to the main paper, as this is the message that the paper is trying to convey?"**
>
> This is a good point. As suggested by the reviewer, space permitting, we will move this from the appendix to the main paper in the final version.
>
> - **"Typos and notation mismatch in the proof of Theorem 1 in the appendix: what is X in eq. 13? index j in eq. 12?"**
>
> Thank you for catching these typos. They are fixed now.
>
> - **"In the final version, it would be also useful to add the filter visualization for the MRI reconstruction as well. It would be interesting to see how dataset dependent the learned filters are."**
>
> This is a great suggestion. As per reviewer’s suggestion, we have included this visualization in Section B.5 of the Appendix.
>
> - **"I am wondering about the complexity of the dual optimization problem. Are there fast convex solvers for scaling up the computations? Also, about the required number of sign patterns, can the authors provide intuitions for the datasets used in the paper, say for MNIST?"**
>
> In our current work, we use SGD for the convex program which is guaranteed to find the optimal solution in linear time, and we use a Pytorch solver that is scalable.
>
> In addition, one can use off-the-shelf convex solvers such as CVXPY to solve convex problems directly, which can be quite parallelized with GPUs.
>
> The primary focus of this work is not to make these convex solvers scalable for these problems, but as stated in Section 6,  this is an important step of our future work.
>
> A remark along these lines is added to the revised paper to clarify the computations, and also the conclusion is updated to include this as an important future work.

---

> ### Comment · AnonReviewer4 · 2020-11-22
>
> Thank you for the response and the new filter visualization for MRI training. I would suggest to include your response about the scalable convex solvers such as PyTorch and CVXPY to the paper.
>
> Considering authors' responses to me and other reviewers, I think, this is a solid paper with sufficient evaluations, thus, I keep my score.

---

### Official Review · AnonReviewer3 · 2020-10-30
**Convex Regularization behind Neural Reconstruction**

**Rating:** 6
**Confidence:** 4

**Review:**

The authors propose a convex formulation for training a 2-layer neural network for reconstruction which should make training easier.

It’s nice to see some results in this domain which are provably convergent, especially since 2 layer NNs are universal approximations. I’m cautiously optimistic that this could open the door to more practically relevant results.

However, the NNs experimented with and the experiments done are relatively simplistic, so it’s hard to grasp how far away this is from practical results. It would also be nice if more effort was spent on contextualising these results within the wider scope of convex methods (e.g. infinite width 2 layer networks).

* The polynomial complexity on its own isn’t that interesting. The input dimension is in the hundreds of thousands, so even quadratic complexity would be highly limiting. It would probably be nice to make it clear that this is mostly of theoretical interest. Perhaps even write out a real estimate for l in the article.
* The main result is quite interesting even outside of regression. Would it be possible to extend it to other convex loss functions?
* Stating that the dual is easier to interpret than the primal is debatable. The primal is also locally linear around almost all points.
* Please make all images in figure 5 equally large.
* Is there a good reason results are not reported on the full FastMRI dataset? It would certainly make comparisons much easier and would contextualize the results. The current setup is not really reproducible.
* Overall I’m missing baselines to understand how relevant the results are. Adding a simple U-Net to the MRI case would disambiguate if this is a relaxation which is of theoretical importance, or if it’s actually giving good results. The images on their own are kinda useless for this.
* The “interpretable reconstruction” part is honestly somewhat debatable. Showing me a few hundred filters might give a tiny bit of insight, but I wouldn’t say it lets me interpret the reconstruction properly.
* Figure 1b is irritatingly not vertically centered
* Images in fig 3 suffer from similar issues. Overall images should be saved in exactly the resolution they are stored in (e.g. 28x28 for MNIST), this is especially important in articles whose whole purpose is to show imaging results. Also avoid linear interpolation between pixels.
* Has the number of sign patterns been ablated?

---

> ### Author Response · Authors · 2020-11-17
> **Response to Reviewer 3**
>
> We would like to thank the reviewer for the comments and suggestions. We have addressed all the reviewer's comments one-by-one below.
>
> - **"The polynomial complexity on its own isn’t that interesting...it would probably be nice to make it clear that this is mostly of theoretical interest."**
>
> Yes, this is of theoretical interest for distinguishing the computational complexity of fully-connected networks from that of fully convolutional networks. We have included a remark on page 4 for clarification. However, the complexity is fully polynomial time for convolutional networks as we clarified in the revised version.
>
> - **"The main result is quite interesting even outside of regression. Would it be possible to extend it to other convex loss functions?"**
>
> Yes. In fact, the results can be extended to every convex loss function. We have included the proof of such an extension in section E of the Appendix.
>
> - **"Stating that the dual is easier to interpret than the primal is debatable. The primal is also locally linear around almost all points."**
>
> While small perturbations in the input will manifest linearly in the output of the primal network (i.e. local linearity), this is quite a different concept than the piecewise linear filtering we illustrate. The primal cannot be expressed in terms of a finite sum of {0, 1}-valued scalar weights applied to a set of linear filters. In contrast, we can do so with the dual, as shown in Figure 1 and eq (8).
>
> - **"Please make all images in figure 5 equally large."**
>
> The updated Figure 5 now has all images with equal size.
>
> - **"Is there a good reason results are not reported on the full FastMRI dataset?"**
>
> A small subset is selected simply to make the experiments tractable. To improve reproducibility, we have included a file “dataset.txt” in our supplemental code which points out to the exact .h5py files from the fastMRI dataset which store the data to be used for training and testing to replicate our results.
>
> - **"Overall I’m missing baselines to understand how relevant the results are. Adding a simple U-Net to the MRI case would disambiguate if this is a relaxation which is of theoretical importance, or if it’s actually giving good results. The images on their own are kinda useless for this."**
>
> The current results we provide are of theoretical importance. One cannot expect a two-layer network to perform as well as a deep U-Net. The two-layer network is simply a setting which can be theoretically analyzed and interpreted. Further work is needed to extend the results to deeper networks. One such way to extend to deeper networks is through greedy layer-wise training, which has shown promising in [Belilovsky et al’2019] and [Nøkland and Eidnes‘2019], but we leave this to future works.
>
> - **"The “interpretable reconstruction” part is honestly somewhat debatable. Showing me a few hundred filters might give a tiny bit of insight, but I wouldn’t say it lets me interpret the reconstruction properly."**
>
> We agree that merely visualizing the filters does not interpret the results. Indeed, one needs to look at the filters along with the patch clustering or association map in Fig. 5 to make interpretation. The visualized filters nicely demonstrate that the network learns a combination of low-pass, band-pass, and high-pass filters. These filters are then selectively applied to individual patches. Thus, combining these, one would be able to find the exact set of filters applied to each input patch of an image for reconstruction.
>
> - **"Figure 1b is irritatingly not vertically centered"**
>
> We have fixed this issue in the revised version--now Fig 1a and 1b are vertically aligned.
>
> - **"Images in fig 3 suffer from similar issues. Overall images should be saved in exactly the resolution they are stored in."**
>
> Thank you for the suggestion. The images are displayed exactly in the same size as they are stored (28x28).
>
> - **"Has the number of sign patterns been ablated?"**
>
> Thank you for the useful suggestion. To address this comment we have included an ablation study, in section B.3 of the Appendix, for the effect of the number of the sign patterns on the quality of the approximation of the dual problem. As shown for the MNIST denoising problem in Figure 10, after randomly subsampling 4,000 sign patterns, for a noise standard deviation of 0.5, the dual problem can suitably represent the solution to the primal problem; whereas when we use a noise standard deviation of 0.1, the dual problem only requires 125 randomly subsampled sign patterns.
>
> **References**
> Eugene Belilovsky, Michael Eickenberg, and Edouard Oyallon. Greedy layerwise learning can scale to imagenet. *In International conference on machine learning*, pp. 583–593. PMLR, 2019.
>
> Arild Nøkland and Lars Hiller Eidnes.   Training neural networks with local error signals.
> *arXiv preprint arXiv:1901.06656*, 2019.

---

### Official Review · AnonReviewer5 · 2020-11-04
**Need more theoretical novelty for ICLR, with doubts for the proof**

**Rating:** 4
**Confidence:** 4

**Review:**

This paper proposes the dual formulation of a two layer neural network, which makes the loss for training convex. The convexity guarantees the global optimality of training step compared to training on the primal loss function. The norm regularizer is applied to ensure the generalization. Then the paper gives explanation and experiments regarding the dual nn.

However, I'm not sure what is the novelty of the work. On the theoretical side, I think Thm 1 is a straightforward computation of the dual function, and I cannot see if there's any novel technique beyond the common textbooks. I think if there's any dual computation in Sec 4.3 for nn with any number of layers, it could be more interesting. So far I don't think it's sufficient for ICLR.

Despite that, even if the majority of contribution is practical performance -- I cannot see what's the goal of the experiments. If this approach is better than the usual nonconvex nn formulation, at least we should see a clear difference. The solid and dash lines are really close to each other, in terms of computation time/epochs or error/loss I cannot see the advantage. The claims in experiment section are vague. E.g., "Zero duality gap", there should be a proof to justify it. I find the proof in appendix eq (12) that $p^* = d^*$. However, to use Slater's condition, is the primal problem convex? I cannot find the proof. (if this is proven I will definitely raise my score) If it's empirical, even the loss looks similar in the experiments, I'm not sure what setup guarantees a zero duality gap (what's the distribution of training samples, what's the underlying relation between x and y, what's the size of nn, etc.) This is really a theoretical claim, otherwise please say "small duality gap observed empirically".

Also it would be great to compare with neural tangent kernel works where GD finds the global optimum of nonconvex primal formulation with high probability. Based on NTK I'm not sure if convexity is so important. Regarding convex NN, this paper is also interesting.
https://openreview.net/forum?id=H1MW72AcK7

=============================== UPDATE ===============================

Checked the proof again and want to confirm that, is it using the composition with pointwise maximum? If so I believe that it's correct, just be more detailed with the proof. Now I'm only concerned with the technical sophistication of the paper. Would be good to discuss deeper NN. Also good to make clear the advantage over other convex or nonconvex but optimizable formulations.

---

> ### Author Response · Authors · 2020-11-17
> **Response to Reviewer 5 (Part 1)**
>
> We would like to thank the reviewer for their comments and suggestions. However, we believe that there are significant misunderstandings, which once clarified can greatly improve this reviewer’s outlook on our paper. Please see our point-to-point response below.
>
> - **"I'm not sure what the novelty of the work is. On the theoretical side, I think Thm 1 is a straightforward computation of the dual function, and I cannot see if there's any novel technique beyond the common textbooks... is it using the composition with pointwise maximum? If so I believe that it's correct, just be more detailed with the proof. "**
>
> We respectfully disagree with the reviewer’s assessment. Theorem 1 is not simply the dual program, it is indeed the ***strong dual***, which is non-trivial to find for this **non-convex** training program.
>
> This powerful claim allows the optimal solution of the convex program (4) to coincide with the optimal solution of the the non-convex neural network training program (2), i.e. $p^* = d^*$.
>
> Our work is the first to show this claim for fully-convolutional two-layer neural networks for image reconstruction problems. This work finds that the computational complexity of solving this fully-convolutional neural network is polynomial in the problem dimensions, which contrasts with previous results on fully-connected networks, for which the training complexity is exponential in the dimension, as we clarified on page 4 in the revised version.
>
> Even for the simple, two-layer network, this result is exceedingly non-trivial and generates new insights into the learned filters of the neural network, and the implicit regularization that is provided by the neural network training objective.
>
> It is clear that our proof requires more steps than a straightforward computation. We demonstrate what a standard duality approach would yield--a dead end--in section F of the Appendix in the revised version. Of particular importance is the re-scaling trick in (9), which allows us to take the dual problem in tractable form. Another novel piece in this proof is the enumeration of sign patterns, which allows us to finitely parameterize the infinite ReLU constraints into $\ell$ constraints using the *pointwise maximum* as the reviewer has noticed, allowing for a finite-dimensional convex program. In this way, the proof of strong duality in Theorem 1 is a non-trivial result.
>
> Accordingly, we believe that these theoretical contributions are sufficiently novel and insightful. To clarify this issue, in the revised version we have detailed the small steps of the proof for Theorem 1. Please see section C of Appendix.
>
> - **"I cannot see what's the goal of the experiments. If this approach is better than the usual nonconvex nn formulation, at least we should see a clear difference. The solid and dash lines are really close to each other, in terms of computation time/epochs or error/loss I cannot see the advantage. The claims in the experiment section are vague."**
>
> The **first point** of the experiments is to verify the theoretical claims of Theorem 1 (i.e. the “zero duality gap”). The point is not to show that dual formulation achieves a better solution than the primal one. That is a promising direction of our convex formulation though, but not the main scope of this submission.
>
> The **second point** of the experiments is to demonstrate the interpretability offered by the dual formulation. The visualization of the dual filters in Fig. 4 is much more interpretable than visualization of primal filters, since the neural network is piecewise linear with respect to the dual filters (as discussed in Section 4.2).

---

> ### Author Response · Authors · 2020-11-17
> **Response to Reviewer 5 (Part 2)**
>
>
> - **"Also it would be great to compare with neural tangent kernel works where GD finds the global optimum of nonconvex primal formulation with high probability. Based on NTK I'm not sure if convexity is so important. Regarding convex NN, this paper is also interesting."**
>
> It is important to clarify that our analyses hold for finite-width networks (number of neurons bounded as per Theorem 1), while NTK relies on the oversimplified infinite-width assumption which simplifies to a linear kernel method.  NTK analysis will not explain the success of practical and finite-width neural networks (see for example Table 1 of [Arora et al’19]).
>
> Indeed, convexity analysis for finite-width networks becomes more tricky, and this is an important contribution of this work. We thank the reviewer for pointing out the interesting paper from Chen et al (2018) relating to Input Convex NNs (ICNN); we have cited it and made clear the distinctions between our work in the revised paper. In particular, while the forward pass of an  ICNN is a convex optimization problem, training ICNNs still requires non-convex optimization, e.g. the authors use SGD heuristically applied to the non-convex training objective.
>
>
> - **"Would be good to discuss deeper NN. "**
>
> Regarding deeper NNs, as commented in section 4.3, a typical trick is to use greedy layer-wise training, where one can sequentially train an unrolled network by stacking shallow networks. This has been found quite promising for training neural networks; see e.g., [Belilovsky et al’2019] and [Nøkland and Eidnes‘2019]. Finding the convex dual for end-to-end trained deep networks is a significantly more challenging problem, as it requires hierarchical groupings of sign patterns which are difficult to characterize. This problem is beyond the scope of our work and can be analyzed in future research. The two-layer problem was already an open-problem to solve, for which the analysis was extensive and non-trivial.
>
> - **"Also good to make clear the advantage over other convex or nonconvex but optimizable formulations."**
>
> The main advantage of this work is that we can **provably** find the global minimum of the non-convex neural network training problem. This contrasts with SGD applied to the non-convex primal training, which is heuristic and may not always find the global minimum, either getting trapped into  a local minimum, or, it converges extremely slowly. For example, refer to Fig. 9 in the Appendix, where under an exponential noise distribution, the primal fails to find the global optimum.
>
> As for other formulations, such as the NTK and ICNNs, both have their own issues. Training an ICNN as mentioned before is not actually a convex optimization problem, while the NTK requires the oversimplified infinite-width assumption, which does not explain the practical finite-width networks.
>
> **References**
> Eugene Belilovsky, Michael Eickenberg, and Edouard Oyallon. Greedy layerwise learning can scale to imagenet. *In International conference on machine learning*, pp. 583–593. PMLR, 2019.
>
> Arild Nøkland and Lars Hiller Eidnes.   Training neural networks with local error signals.
> *arXiv preprint arXiv:1901.06656*, 2019.
>
> Yize Chen,  Yuanyuan Shi,  and Baosen Zhang.   Optimal control via neural networks:  A convex approach.   *In International Conference on Learning Representations*, 2019.
>
> Sanjeev  Arora,  Simon  S  Du,  Wei  Hu,  Zhiyuan  Li,  Russ  R  Salakhutdinov,  and  Ruosong  Wang. On exact computation with an infinitely wide neural net. *In Advances in Neural Information Processing Systems*, pp. 8141–8150, 2019.

---

### Author Response · Authors · 2020-11-17
**Summary of Responses to Reviewers**

We thank the reviewers for their helpful comments and suggestions. We have addressed all the comments one-by-one, which we believe has improved the paper. To track the changes, they are marked in blue in the revised version. The major changes are also summarized below:

- In accordance with R5’s comments, the proof of Theorem 1 is detailed, and each single step is elaborated.
- Following the suggestion from R3, an ablation study is included for the number of sign patterns for MNIST denoising
- Per R4's comment, we have included further experiments for the MRI filter visualization.
- The differences between our work and the related ones suggested by R2 are discussed. Those include the NTK analysis, [Papyan et al’2017], [Yokota et al’2019], and [Tachella et al’2020].
- Several clarifying statements are included and the typographical errors as well as issues with figures are resolved.

---

### Decision · Program_Chairs · 2021-01-07
**Final Decision**

**Decision:**

Accept (Poster)

**Comment:**

This paper is motivated by figuring out what regularization do popular neural network reconstruction techniques correspond to. In particular, this paper studies a convex duality framework that characterizes the global optima of a two-layer fully-convolutional ReLU denoising network via convex optimization. The authors use this regularization to interpret the obtained training results. The reviewers raised a variety of concerns regarding the tractability of the optimization problem (seems to be exponential in number of constraints), the utility for interpretation etc, significance of the results compared to existing literature. Some of these concerns were alleviated but not fully resolved. One reviewer had concerns about the correctness of the proof that was resolved based on the authors’ response. I share many of the above concerns. However, I do think having a computationally feasible way to figure out the exact regularization in these simple settings (at least with small dimensions) could provide some insights to guide further theoretical development.  Therefore I am recommending acceptance. However, I strongly urge the authors to further revise the paper based on the above comments.